# Species-Specific Response of Corals to Imbalanced Ratios of Inorganic Nutrients

**DOI:** 10.3390/ijms24043119

**Published:** 2023-02-04

**Authors:** Alice C. A. Blanckaert, Tom Biscéré, Renaud Grover, Christine Ferrier-Pagès

**Affiliations:** 1Coral Ecophysiology Team, Centre Scientifique de Monaco, 8 Quai Antoine 1er, MC-98000 Monaco, Monaco; 2IFD-ED 129, Sorbonne Université Sciences (Formerly UPMC Université Paris VI), CEDEX 05, 75005 Paris, France

**Keywords:** DIN, DIP, octocoral, scleractinian, elemental composition, nutrient uptake

## Abstract

Dissolved inorganic phosphorus (DIP) is a limiting nutrient in the physiology of scleractinian corals. Anthropogenic addition of dissolved inorganic nitrogen (DIN) to coastal reefs increases the seawater DIN:DIP ratio and further increases P limitation, which is detrimental to coral health. The effects of imbalanced DIN:DIP ratios on coral physiology require further investigation in coral species other than the most studied branching corals. Here we investigated the nutrient uptake rates, elemental tissue composition and physiology of a foliose stony coral, *Turbinaria reniformis*, and a soft coral, *Sarcophyton glaucum*, exposed to four different DIN: DIP ratios (0.5:0.2, 0.5:1, 3:0.2, 3:1). The results show that *T. reniformis* had high uptake rates of DIN and DIP, proportional to the seawater nutrient concentrations. DIN enrichment alone led to an increase in tissue N content, shifting the tissue N:P ratio towards P limitation. However, *S. glaucum* had 5 times lower uptake rates and only took up DIN when the seawater was simultaneously enriched with DIP. This double uptake of N and P did not alter tissue stoichiometry. This study allows us to better understand the susceptibility of corals to changes in the DIN:DIP ratio and predict how coral species will respond under eutrophic conditions in the reef.

## 1. Introduction

Coral reefs are one of the most diverse and productive ecosystems in the world, even though they thrive in oligotrophic waters that have low levels of dissolved inorganic nitrogen (DIN, <0.5 μM) and phosphorus (DIP, <0.2 μM). This paradox, commonly referred to as the “Darwin Paradox” [1], is explained by a highly efficient nutrient cycling between corals, which are the building blocks of the reefs, and their symbiotic microorganisms, including Symbiodiniaceae dinoflagellates [2,3]. Dinoflagellates metabolize DIN and DIP from seawater or from the host’s metabolic waste products into organic nutrients, which are then translocated to the coral host to meet most of its energy needs [4]. In turn, the host captures planktonic prey and shares this nutrient source with its dinoflagellates [5]. Finally, other coral-associated microbes, such as bacteria and fungi, can potentially be involved in carbon fixation, nitrogen metabolism and sulfur cycling, among many other functions [3].

The close nutritional relationship between corals and Symbiodiniaceae responds strongly to the nutrient environment. On the one hand, nutrient deficiencies, especially in phosphorus, which is often a limiting nutrient in reef waters [6,7], impair coral health and exacerbate coral bleaching (i.e., the loss of symbionts from coral host tissues) under heat stress conditions [8,9,10]. On the other hand, experimental exposure to elevated seawater nutrient concentrations, such as those found in reefs exposed to agricultural and urban wastes [11,12], may also affect coral physiology (Appendix A). Studies have shown that calcification rates, skeletal density, photosynthate fluxes or reproductive success are reduced under nitrate [13,14,15,16,17,18,19] or phosphate enrichment [15,20,21,22]. However, this impairment in coral physiology under high concentrations of dissolved inorganic nutrients is not systematic, as in situ and laboratory observations have also reported healthy corals exposed to high nutrient concentrations [17,18,23,24,25]. In some cases, nutrient enrichment may even improve coral growth or reduce their susceptibility to bleaching [16,17,19,26,27,28,29,30,31]. Wiedenmann et al. [16] were the first to suggest that it is not the nutrient concentration per se but rather an imbalanced DIN:DIP ratio that is detrimental to coral physiology. DIN:DIP ratios are considered balanced between 3 and 7, which are the ratios commonly recorded in pristine reefs [10,32]. Significant deviations from these values might indicate potential nutrient limitation, with low DIN:DIP ratios suggesting nitrogen limitation and high DIN:DIP ratios suggesting phosphorus limitation. 

The observed physiological changes of corals exposed to imbalanced DIN:DIP ratios are not well explained. A high DIN:DIP ratio (nitrogen enrichment) was shown to enhance symbiont growth and induce phosphorus starvation. In turn, phosphorus deficiency reduces the photosynthetic capacity of corals and can impair the structure of the phospholipid membranes of the symbionts, lowering the temperature thresholds for coral bleaching [7,8,10,33]. Therefore, imbalanced DIN:DIP ratios can induce significant changes in the stoichiometry (Carbon (C):N:P) of coral tissue, itself dependent on DIN and DIP availability in seawater, as observed in phytoplankton [34,35]. Because P is essential for ribosomal RNA and N is a major component of nucleic acids and proteins, coral growth and health are directly related to tissue N and P content [36,37]. Measuring the elemental stoichiometry of coral tissues and symbionts may therefore be used to determine nutrient limitation and assess how the seawater DIN:DIP ratio might alter coral physiology. However, to date, changes in coral tissue composition with varying DIN:DIP ratios have been poorly studied. Of the 47 studies reported in Appendix A on the effects of nutrient enrichment and imbalanced DIN:DIP ratios on coral physiology, only four studies measured the C:N content of coral tissue and symbionts under N enrichment (ammonium: [38,39,40], and nitrate: [41]) and three under combined N and P enrichment [28,41,42] (Appendix A). The N:P or C:P ratio of coral tissue and symbionts was not considered, although P can be an important limiting compound for coral physiology [8]. In addition, most studies focused on branching coral species (*Acropora, Stylophora* and *Pocillopora*), while other coral morphologies or groups were hardly studied (Appendix A). 

This study therefore investigated the effect of balanced and imbalanced DIN:DIP ratios on the physiology and elemental stoichiometry of two understudied coral groups: the scleractinian coral *Turbinaria reniformis*, which is a foliose coral, and the octocoral *Sarcophyton glaucum*. These species are different from scleractinian corals, which are particularly abundant in oligotrophic, pristine environments. *T. reniformis* is indeed known to thrive in turbid, particle-rich environments [43], while *S. glaucum* belongs to the octocorals, known to dominate the benthic community in nutrient-rich environments [44]. It addresses two questions. First, are there different effects of balanced and imbalanced DIN:DIP ratios on the nutrient uptake rates, physiology, and cellular C:N:P stoichiometry of the corals? Second, do the scleractinian coral and the octocoral respond in the same way to changes in DIN:DIP ratios? We hypothesize that an imbalanced DIN:DIP ratio should be less detrimental to octocoral physiology, as octocorals appear to be dominant over scleractinian corals in eutrophic reefs [44]. To answer these questions, corals were exposed to two balanced DIN:DIP ratios with different nutrient concentrations: DIN:DIP_2.5_ (0.5:0.2) and DIN:DIP_3_ (3:1) and two imbalanced DIN:DIP ratios: DIN:DIP_0.5_ (0.5:1) and DIN:DIP_15_ (3:0.2). The DIN:DIP_2.5_ corresponds to the natural seawater control condition and will be referred to as the control condition in the results and discussion sections. The results of this study will provide useful insight on the future of corals submitted to eutrophication and inform on the different managerial strategies that can be implemented depending on the dominant coral species.

## 2. Results

### 2.1. Effect of DIN:DIP Ratios on the Physiology and Tissue Composition of Turbinaria Reniformis

Significant changes in *T. reniformis* physiology occurred primarily under the balanced DIN:DIP_3_ (combined enrichment in DIN and DIP) and are presented first in this result section (Appendix A, for median and standard error values; Appendix A for statistical results). This nutrient condition increased the rates of DIN (nitrate) and DIP (phosphate) uptake by fifteen (Figure 1a, *p* value 0.001) and twenty-fold (Figure 1b, *p* value 0.001), respectively, compared to the control condition, DIN:DIP_2.5_. Although both the density of symbiotic dinoflagellates (Figure 2a, *p* value 0.06) and the protein content (Figure 2b, *p* value 0.4) remained unchanged, the chlorophyll a and c_2_ content (Figure 3, *p* value 0.02) and net photosynthetic rates (Figure 4a, *p* value 0.002) doubled compared to the control condition as a result of higher nutrient availability. This increase in carbon assimilation was followed by an increase in carbon reserves with a doubling of lipid (Figure 5a, *p* value 0.03) and carbohydrate content (Figure 5b, *p* value 0.01) as well as an increase in the total carbon content (Figure 6a, *p* value 0.05). Moreover, due to increased nitrate, there was a subsequent increase in the total nitrogen content of *T. reniformis* (Figure 6b, *p* value 0.04), so that the C:N ratio remained unchanged compared to the control condition (Figure 6c, *p* value 0.5). On the other hand, despite enhanced DIP uptake under DIN:DIP_3_, the phosphorus content remained unchanged (Figure 7a, *p* value 0.15), likely due to the large variability in the phosphorus content in the control condition. Therefore, the C:P ratio was not statistically different from the control level, although it almost doubled on average (Figure 7b, *p* value 0.06). 

The imbalance in DIN:DIP_15_ (DIN enrichment alone) significantly increased nitrate uptake rates in *T. reniformis* by sevenfold (Figure 1a, *p* value 0.005) and thus the nitrogen content in coral tissue (Figure 6b, *p* value 0.03) compared to the control condition. As a result, the cellular N:P ratio under the imbalanced DIN:DIP_15_ was twice that of the control condition (Figure 7c, *p* value 0.01). Although some other parameters (chlorophyll content, net photosynthetic rates, carbohydrate, lipid and carbon content) tended to increase with DIN:DIP_15_, this increase was not significant (Figure 3, Figure 4a, Figure 5 and Figure 6a, *p* values > 0.1). The other physiological parameters (P content, C:N and C:P ratios, symbiont density and protein content) remained unchanged compared to the control condition (Figure 2, Figure 6c, and Figure 7a,b, *p* value > 0.05). 

The imbalanced DIN:DIP_0.5_ (DIP enrichment alone) enhanced the uptake rates of phosphate (Figure 1b, *p* value 0.001) and reduced the symbiotic dinoflagellate density by half (Figure 2b, *p* value 0.001), without any change in total chlorophyll content and photosynthetic rates (Figure 3 and Figure 4a, *p* value > 0.6). On the other hand, DIN:DIP_0.5_ doubled the carbohydrate content (Figure 5a, *p* value 0.01) as well as the total carbon (Figure 6a, *p* value 0.003) and nitrogen content (Figure 6b, *p* value 0.01) compared to the control condition. However, the P content remained unchanged despite a tendency to increase (Figure 7a, *p* value 0.15). Both the C:P and N:P ratios also remained unaffected in DIN:DIP_0.5_ compared to the control, likely due to the variability in P content among samples (Figure 7b,c, *p* values > 0.7).

### 2.2. Effect of DIN:DIP Ratios on the Physiology and Tissue Composition of Sarcophyton Glaucum

Overall, the different DIN:DIP ratios had a limited effect on *S. glaucum* physiology (Appendix A). The nitrate uptake rate doubled in DIN:DIP_3_ (DIN and DIP enrichment) compared to the control condition (Figure 1a, *p* value 0.02), while the phosphate uptake rate remained unchanged (Figure 1b, *p* value 0.46). In addition, carbohydrate content decreased by half under DIN:DIP_15_ (DIN enrichment) compared to the control condition DIN:DIP_2.5_ (Figure 5b, *p* value 0.004). However, under DIN:DIP_0.5_ (DIP enrichment), the carbohydrate content increased (Figure 5a, *p* value 0.02) along with the protein content (Figure 2b, *p* value 0.01). The other parameters remained at the control level under the different balanced and imbalanced DIN:DIP ratios (Figure 3, Figure 4, Figure 5, Figure 6 and Figure 7, *p* values > 0.2).

### 2.3. Comparison in the Physiological Parameters between Turbinaria Reniformis and Sarcophyton Glaucum under a Control Balanced DIN:DIP_2.5_ without Enrichment

Despite similar dinoflagellate density (Figure 2b, *p* value 0.14) and total chlorophyll content (Figure 3, *p* value 0.7) in both coral species, the net photosynthetic rate and respiration rate of *S. glaucum* were significantly lower than those of *T. reniformis* (Figure 4a,b, *p* values 0.002). While carbohydrate content was similar between coral species (Figure 5a, *p* value 0.06), lipid content was ten times higher in *S. glaucum* than in *T. reniformis* (Figure 5b, *p* value 0.001). However, the protein content was divided by two in *S. glaucum* (Figure 2a, *p* value 0.002). Overall, the carbon (Figure 6a, *p* value 0.03) and nitrogen (Figure 6b, *p* value 0.03) contents were significantly higher in *S. glaucum* than in *T. reniformis*, while the phosphorus content was similar in both coral species (Figure 7a, *p* value 0.85). Thus, the C:N (Figure 6c, *p* value 0.03), C:P (Figure 7b, *p* value 0.03) and N:P (Figure 7c, *p* value 0.004) ratios were significantly higher in *S. glaucum* than in *T. reniformis*.

## 3. Discussion

Declining water quality in many reefs is causing phase shifts in the composition of the benthic community, from a dominance of scleractinian corals to a dominance of macroalgae [45] and other benthic organisms such as soft corals [44], which may therefore be the dominant benthic organisms in the future. Numerous studies have therefore examined the effects of organic and inorganic nutrient pollution on scleractinian corals, which are the main reef builders. These studies have shown that both types of pollution affect the metabolism and resistance of scleractinian corals to thermal stress [16,19,46,47]. This is because nutrient enrichment can lead to the disruption of the symbiosis between corals and their associated microbiota [47,48,49]. In contrast to scleractinian corals, studies have shown that organic and inorganic contaminants do not always have detrimental effects on soft corals or even increase their resistance to thermal stress [22,50], which may facilitate their ecological dominance over scleractinian corals. Therefore, further studies comparing the responses of soft corals and scleractinian corals to nutrient enrichment are needed to better predict future reef development. The results of this study show that the physiology and tissue stoichiometry of *S. glaucum* did not change under the different DIN:DIP ratio scenarios because this species, as many soft corals, had very low nutrient uptake rates. Therefore, little N and P entered the symbiosis at varying DIN:DIP ratios. In contrast, *T. reniformis*, as all hard corals, took up nutrients proportionally to their concentration in seawater. The massive input of N under nitrogen-imbalanced conditions (DIN:DIP_15_) led to significant changes in tissue stoichiometry and may explain the impairment of coral physiology observed in many previous studies. However, a balanced enrichment in DIN and DIP (DIN:DIP_3_) resulted in improved physiology and metabolism of *T. reniformis*, suggesting that the physiology of scleractinian corals is co-limited by N and P. 

### 3.1. Effect of DIN:DIP Ratio on the Physiology of T. Reniformis

The results of this study first show that changes in seawater DIN:DIP ratios can induce changes in nutrient uptake rates by the scleractinian coral *T. reniformis*. Uptake of DIN and DIP at different DIN:DIP ratios has been poorly studied in photosynthetic organisms and never with coral species to our knowledge (see [51,52] for macroalgae, [53] for cyanobacteria, and [54] for trees). This study first confirms the findings of previous studies that uptake rates are dependent on the nutrient concentration in seawater, with increased uptake rates at high nutrient concentrations (DIN:DIP_3_) [51,52,53]. Furthermore, our results demonstrate that the availability of one nutrient increases the uptake rate of the other nutrient, as has been observed in phytoplankton species but not in corals [53]. The highest uptake rates of DIN were indeed obtained with combined enrichment (DIN:DIP_3_). In contrast, *T. reniformis* took up DIP whenever its concentration in seawater was elevated (DIN:DIP_3_ and DIN:DIP_0.5_), indicating that P was a limiting nutrient for this coral species under all conditions. Overall, P is known to be an essential nutrient for corals [8,55] and is found in limited concentrations in the environment. A deficiency in P leads to alteration of the thylakoid membranes of the symbiont and increases susceptibility to bleaching [16,56]. 

In this study, two conditions (DIN:DIP_0.5_ and DIN:DIP_15_) were imbalanced compared to what is normally observed in reefs (DIN:DIP between 4 and 7: [10,32]). Such a situation may induce a subsequent change in element concentrations and stoichiometry in the coral tissue, and lead to imbalanced cellular N:P ratios. A significant effect of imbalanced DIN:DIP ratios was indeed observed on the physiology and/or elemental stoichiometry of the scleractinian coral *T. reniformis*, depending on whether seawater was enriched with DIP or DIN. Overall, DIP enrichment (DIN:DIP_0.5_) increased the total carbon (as carbohydrates) and nitrogen content of coral tissue without altering the internal elemental stoichiometry. These results are consistent with the notion that the availability of a limiting nutrient (especially phosphorus) promotes the uptake and/or assimilation of the other nutrients [52,53]. This is also consistent with a previous study showing that the carbon content in *Montipora digitata* symbionts increased significantly when DIP was increased in seawater [41]. On the contrary, enrichment with DIN alone (DIN:DIP_15_) did not enhance the production of organic carbon or the assimilation of phosphorus, neither in this study nor in the previous study by Tanaka et al. [41]. Instead, enrichment with DIN significantly increased the N content and the N:P ratio in the coral tissue, as previously observed by Wiedenmann et al. [16]. In this study, the N:P ratio increased from about 20 in the control condition (balanced DIN:DIP_2.5_) to about 60 in the imbalanced DIN:DIP_15_. Such a situation is known to be detrimental to coral growth and health in many branching species [14,17,19,22,28], in particular under thermal stress [16]. No physiological changes were measured in *T. reniformis* during this relatively short-term incubation with DIN enrichment, as previously observed [25,57,58]. The results might have been different for a long-term enrichment incubation or if a thermal stress had been applied. On the contrary, other scleractinian species showed an impairment of their physiology after the same incubation time with similar nitrogen concentrations [16,19]. All together, the results obtained suggest that *T. reniformis* belongs to the scleractinian species that may be the most resistant to changes in seawater stoichiometry. Accordingly, Tanaka et al. [41] observed species-specific susceptibility to unbalanced DIN:DIP ratios. The stronger effects were observed in the symbionts of *M. digitata* compared to *Porites cylindrica*. The authors hypothesized that this was due to the lower N content per skeletal surface of *M. digitata*, making the symbionts more dependent on seawater and host-derived N [41]. 

In contrast to the imbalanced DIN:DIP conditions, increased DIN and DIP concentrations with a balanced ratio (DIN:DIP_3_) did not change the cellular stoichiometry of T. reniformis because uptake rates increased for both nutrients. All measured physiological and tissue parameters were even increased under this balanced nutrient enrichment. Overall, given the significant increase in net and gross photosynthetic rates, as well as in carbohydrate and lipid content, the dual enrichment can be considered beneficial to coral physiological performances, as previously suggested [25,58,59]. Although calcification rates were not assessed in this study, moderate nutrient enrichment, as used in this study, has been shown to be beneficial to corals by mitigating the negative effects of thermal stress or acidification [8,9,30,60,61] without negatively affecting calcification [28,62,63]. Although there is a beneficial direct effect of DIN and DIP enrichment on coral performance, we must keep in mind that many indirect effects, such as increased bacterial and phytoplankton populations, macroalgal growth and corallivorous predators, might occur under natural reef conditions [7]. 

### 3.2. Effect of DIN:DIP Ratio on the Physiology of S. Glaucum

The soft coral species *S. glaucum* responded differently than *T. reniformis* to a change in the DIN:DIP ratio. First of all, *S. glaucum* exhibited much lower nitrogen and phosphorus uptake rates than T. reniformis, even under high nutrient concentrations in seawater (DIN:DIP_3_). These results are in agreement with a previous study [64], which found lower nutrient uptake and assimilation rates in soft corals compared to scleractinian corals. Several reasons were proposed for this. Different Symbiodiniaceae species and densities may have different nutrient acquisition efficiencies [64]. The coral microbiome can also lead to variations in nutrient supply, for example, through the presence of diazotrophic bacteria in scleractinian corals [65,66]. Morphological characteristics may also explain the observed differences in nutrient uptake. In soft corals, the presence of a thick mesoglea may act like a barrier and prevent the diffusion of molecules such as DIC [67]. On the other hand, in scleractinian corals, the presence of a calcium carbonate skeleton generates protons [68], which are buffered in the presence of ammonium [69,70]. Nutrient uptake in corals is also enhanced in the light by photosynthesis [71]. Thus, the lower nutrient uptake rate in soft corals could be explained by their consistently lower photosynthetic rates compared to scleractinian corals, as observed in both this study and by Pupier et al. [66]. It might also be due to the thick mesoglea and tissue layer of soft corals, which may act as a barrier to the diffusion of inorganic nutrients. The low nutrient uptake rates in *S. glaucum* compared to *T. reniformis* may induce a continuous P and N limitation in this species and may also explain the higher tissue C:N:P ratio. While the N:P ratio in *T. reniformis* is slightly higher than the Redfield ratio (20 vs. 16), it indeed reached up to 80 in *S. glaucum*. The high carbon content in *S. glaucum* is evidenced by its 10 times higher lipid content (per mg AFDW) than that in *T. reniformis*. In contrast, the protein content, requiring N and P, is two times lower in *S. glaucum*. However, it should be noted that for *S. glaucum*, traces of calcium carbonate spicules could also drive the measured high carbon content in the tissues.

The low nutrient uptake rates of *S. glaucum* may explain why no major changes in tissue stoichiometry (C:N:P) or element concentrations were observed under imbalanced DIN:DIP ratios. Low nutrient uptake rates may have prevented a massive input of nutrients into the coral tissue and subsequent changes in host-symbiont interactions or elemental stoichiometry. Under a combined and balanced DIN and DIP enrichment (DIN:DIP_3_), there was a significant increase in nutrient uptake rates, albeit relatively low compared to T. reniformis. As a consequence, no effects on S. glaucum physiology, element concentration or tissue stoichiometry were observed. 

The lack of large physiological changes under imbalanced or enriched DIN:DIP conditions could explain why soft corals are less affected by reef nutrification than scleractinian corals [44,72]. Future studies should aim at understanding the effects of imbalanced DIN:DIP ratios on the trophic food chain and the consequences for coral nutrition. Soft corals are indeed supposed to rely more on heterotrophy and phytoplankton grazing than scleractinian corals [73,74,75]. Therefore, any increase in seawater DIN or DIP may promote phytoplankton growth, which in turn benefits the growth of soft corals through increased feeding. 

All together, these results suggest that soft corals such as S. glaucum, which have low nutrient uptake rates, can be relatively unaffected by imbalanced DIN:DIP ratios of the seawater because nutrients do not massively enter the coral tissue and do not lead to imbalanced cellular elemental ratios. On the contrary, species with high uptake rates, such as all scleractinian corals, will be more affected by imbalanced DIN:DIP ratios in the seawater. Indeed, they will experience a large influx of nutrients within the coral tissue, which will lead to imbalanced cellular elemental ratios. The response to imbalanced DIN:DIP ratios is however species-specific in scleractinian corals. 

## 4. Materials and Methods

### 4.1. Experimental Design

Two widespread Red Sea coral species were used for this experiment, the octocoral *Sarcophyton glaucum* and the scleractinian coral *Turbinaria reniformis*. These two species were maintained under oligotrophic, nutrient-poor conditions, similar to the Red Sea reef conditions in shallow waters. Therefore, total dissolved inorganic nitrogen (DIN) and phosphorus (DIP) concentrations were around 0.5 µM and 0.2 µM [76]. In the Red Sea, these concentrations remain low throughout the years in shallow waters [76]. For each coral species, 96 nubbins were produced from 12 different colonies (eight nubbins per colony) and then equally divided into eight aquaria (one nubbin per colony, 12 nubbins per species and aquaria). Nubbins were left for three weeks in an open system continuously supplied with filtered seawater, pumped at 40m depth and illuminated by 400W metal halide lamps (HPI-T, Philips, Amsterdam, The Netherlands) at an irradiance of 200 ± 10 µmol photons m^−2^ s^−1^ (12h:12h photoperiod). The temperature was controlled at 25 °C by heaters connected to Elli-Well PC 902/T controllers. Nubbins were fed twice a week with *Artemia salina* nauplii to help with coral healing. After healing, feeding was stopped, and nubbins were maintained in this condition for a week before the following experimental design was applied. 

The eight aquaria were divided into four sets of two aquaria, with each set maintained at a different concentration of dissolved inorganic nitrogen (DIN) and phosphorus (DIP) (Appendix A). The first set was the control condition with a balanced DIN:DIP ratio of 2.5 (named “Control, balanced DIN:DIP_2.5_”), which received natural seawater (undetectable amounts of ammonium, 0.5 μM nitrate and 0.2 μM phosphate). The second set of aquaria with an imbalanced DIN:DIP ratio of 0.5 (named “DIN:DIP_0.5_”) was enriched with 1 µM DIP delivered in the form of phosphates. The third set of aquaria with an imbalanced DIN:DIP ratio of 15 (named “DIN:DIP_15_”) was enriched with 3 µM DIN delivered in the form of nitrate. Finally, the last set of aquaria with a balanced DIN:DIP ratio of 3 (named “DIN:DIP_3_”) was enriched with both 3 µM nitrate and 1 µM phosphate to reach an equivalent DIN:DIP ratio to the control condition, DIN:DIP_2.5_. These concentrations were chosen as they are commonly measured in reefs under agricultural and urban pollution [77]. DIN and DIP were delivered to each aquarium by two peristaltic pumps (Ismatec) from two concentrated stock solutions of sodium nitrate (NaNO_3_) and di-sodium phosphate (Na_2_HPO_4_). To ensure water mixing and a homogeneous nutrient concentration, submersible pumps (Aquarium system, micro-jet MC 320, Mentor, OH, USA) were set up in each aquarium. Nutrient concentrations were controlled using a 3 HR Autoanalyzer (SEAL Analytical, Mequon, WI, USA) according to Aminot et al. [78]. Nubbins were maintained in these conditions for five weeks. Four sets of measurements were performed with six to four nubbins each, sampled from different colonies and different aquaria (Appendix A). Six nubbins were used for the measurements of nutrient uptake rates, photosynthetic efficiency, oxygen fluxes, symbiont density, chlorophyll and protein concentrations. A second set of six nubbins was used for the determination of the lipid and carbohydrate content. Oxidative stress (lipid peroxidation and antioxidant capacity) was obtained from the third set of six nubbins, while the C,N,P tissue content was measured on a fourth set of six nubbins.

### 4.2. Dissolved Inorganic Nutrients Uptake Rate

Nutrient uptake rates under different DIN:DIP ratios were measured on six nubbins per condition. Nubbins were incubated in individual beakers filled with 200 mL of 0.45 μm filtered seawater, maintained at 25 °C and stirred. No DIN and DIP were added to the control beakers. On the contrary, 3 µM of nitrate were added to the DIN:DIP_15_ and DIN:DIP_3_ conditions from a solution of sodium nitrate and 1 μM of phosphate was added to the conditions DIN:DIP_0.5_ and DIN:DIP_3_ from a solution of disodium phosphate. Immediately after the addition of nitrate and phosphate to the respective beakers and every hour for three hours, subsamples of 10 mL of seawater were taken. They were filtered through a 0.45 μm filter syringe in a 15-mL Falcon tube and stored at 4 °C until analysis the following day. Nutrient concentrations in the seawater subsamples were analyzed using a 3 HR Autoanalyzer (SEAL Analytical, USA) according to Aminot et al. [78]. Nitrate and phosphate uptake rates under the different conditions were measured by their depletion in the seawater between the different timepoints. 

### 4.3. Measurement of Photosynthetic Parameters

Rates of respiration (R) and net photosynthesis (Pn) were measured on the same six nubbins as in 4.2. Nubbins were transferred into a 60 mL closed transparent plexiglass chamber filled with 0.45 μm filtered seawater, maintained at 25 °C and stirred. Chambers were equipped with an oxygen sensor (Polymere Optical Fiber, PreSens, Regensburg, Germany) connected by an optical fiber to an Oxy-4 (Channel fiber-optic oxygen meter, PreSens, Regensburg, Germany). The oxygen concentration was recorded with the Oxy4v2-30fb software, in the dark for R and at 200 µmol photons m^−2^ s^−1^ for Pn. Calibrations were conducted at 0% O_2_ with nitrogen-saturated seawater and at 100% O_2_ with air-saturated seawater. Incubations were stopped when a variation of at least 10% in the dissolved oxygen concentration was reached. The gross photosynthesis (Pg) rate was obtained by adding Pn and the absolute value of R. This calculated Pg is likely to be an underestimation of the actual gross photosynthesis rate [79]. Nubbins were used for the nutrient uptake rate measurements and then frozen at −80 °C for later analysis of tissue parameters.

The efficiency of the photosystem II of the dinoflagellate (Fv/Fm) was also measured on the same six nubbins. The nubbins were first placed in darkness for 10 min. Then, using an optical fiber connected to a Dual Pam/F (Fiber version, Heinz Waltz, Germany), a saturated pulse of photosynthetically active radiation (PAR) was sent and the Fv/Fm recorded. Nubbins were not sacrificed after this measurement and used for the different analyses.

### 4.4. Tissue Parameters

All data were normalized to ash-free dry weight (AFDW), according to Pupier et al. [80], or protein biomass, since skeletal surface area was impossible to retrieve from soft corals. After being sampled, *S. glaucum* nubbins were directly lyophilized for 72 h (Christ Martin™ Alpha™ Freeze dryer, Fisher Scientific, Hampton, NH, USA), while *T. reniformis* nubbins were first water-picked to remove the tissue from the skeleton before being lyophilized. The dry weight of the nubbins was then recorded, and the AFDW was determined by burning a known weight of powder at 450 °C for 4 h and subtracting the ash weight (inorganic fraction) from the dry weight. 

#### 4.4.1. Symbiodinium Density, Total Protein and Chlorophyll Content

These three parameters were assessed from nubbins used for the photosynthetic measurements. Freeze-dried samples were re-suspended in 10 mL of filtered seawater and homogenized with a Potter tissue grinder. Freeze-drying the samples did not impact the measurement of these three parameters [80]. From each nubbin, a 5 mL sub-sample was used for the determination of the total chlorophyll a and c_2_ concentration. It was first centrifuged at 3000× *g* for 10 min at 4 °C to pellet the dinoflagellates. The pellet was then re-suspended in 5 mL of 100% acetone to extract chlorophyll a and c_2_ at 4°C in the dark for 24 h. Finally, the extract was centrifuged at 15 °C, 530 g for 15 min before reading the absorbance at 630, 663 and 750 nm with a spectrofluorometer (UVmc_2_ Safas, Xenius, Monaco). Chlorophyll a and c_2_ concentrations were calculated according to the equations of Jeffrey and Humphrey [81].

The protein content of the coral holobiont was quantified in a 500-μL subsample of each nubbin. The subsample was incubated for 5 h at 60 °C in 0.5 M sodium hydroxide (1:1). It was then centrifuged for 1 min at 1000× *g* and the supernatant was distributed in a 96-well microplate in triplicate. The bicinchoninic acid (BCA) assay kit solution (Interchim) was then added to each well according to Smith et al. [82] before incubating the microplate for 30 min at 60 °C. Finally, the absorbance was read at 562 nm with a spectrofluorometer (UVmc2 SAFAS, Xenius, Monaco). The standard curve was conducted with known concentrations of bovine serum albumin (BSA). Finally, a 100 μL subsample from each nubbin was used to estimate the dinoflagellate density with a Z1 Coulter Particle Counter (Beckman Coulter, Pasadena, CA, USA). Each sample was counted in triplicate, and each measurement was performed twice.

#### 4.4.2. Lipid and Carbohydrate Content

Six nubbins per treatment and species were flash frozen in liquid nitrogen and lyophilized for the subsequent measurement of lipid and carbohydrate content. Holobiont lipid content was measured using pre-burned (450 °C for 5 h) Pyrex glass tubes and the phospho-sulpho-vanillin reaction as described in Barnes and Blackstock [83]. Briefly, from each nubbin, a 20 mg sub-sample was suspended in a solution of chloroform/methanol (2:1), mixed for 20 min, and then centrifuged at 3000× *g* for 5 min at 4 °C. The supernatant was collected in a new Pyrex glass tube and placed in a dry bath at 90 °C until complete evaporation. The residue was treated with 500 μL concentrated sulfuric acid before adding vanillin reagent for 30 min in the dark. Absorbance was then read at 520 nm with a spectrofluorometer (Xenius^®^, SAFAS, Monaco). The standard curve was prepared with known concentrations of cholesterol [84].

The carbohydrate content was assessed using the Total Carbohydrate Assay kit (Sigma Aldrich, St. Louis, MO, USA), based on the phenol-sulfuric acid method, following the manufacturer’s instructions. Briefly, from each nubbin, a sub-sample was sonicated (20 pulses at 70Hz, Vibra-Cell™, Bioblock Scientific, Illkirch, France) on ice in cold 1X PBS (Phosphate-Buffered Saline), then centrifuged at 10,000× *g* for 5 min. The supernatant was deposited in a 96-well plate with sulfuric acid and incubated in the dark for 15 min at 90 °C before receiving the developer solution from the manufacturer’s kit. After 5 min, the absorbance was read at 490 nm with a spectrofluorometer (Xenius^®^, SAFAS, Monaco). The standard curve was prepared with known concentrations of D-glucose.

### 4.5. Oxidative Status

Oxidative stress analysis was performed on six nubbins per condition and species, snap-frozen in liquid nitrogen and kept at −80 °C before analysis. Each analysis was performed on a 0.5–1 cm fragment cut from the main nubbin.

#### 4.5.1. Non-Enzymatic Total Antioxidant Capacity (NETAC)

Fragments were sonicated on ice in cold PBS 1X and centrifuged at 10,000× *g* for 10 min at 4 °C. Protein content was measured as described above. All samples were diluted to the same protein concentration before NETAC was assessed with the OxiSelectTM Total Antioxidant Capacity Assay kit (Cell Biolabs Inc., San Diego, CA, USA), based on the reduction of copper (II) to copper (I) by antioxidants. Final samples were added to a 96-well plate with the reaction buffer, and the absorbance was read at 490 nm with a spectrofluorometer (Xenius^®^, SAFAS, Monaco). Results were expressed as µM copper reducing equivalents (CRE). mg protein^−1^. The standard curve was prepared with known concentrations of uric acid.

#### 4.5.2. Lipid Peroxidation (LPO)

Fragments were sonicated on ice in a homogenizing buffer composed of a KCL solution (1.15%) with 35 μM butylated hydroxytoluene (BHT). Samples were then centrifuged at 10,000× *g* at 4 °C for 10 min and protein content was measured as described above and standardized to 0.8 μg μL^−1^. The lipid peroxidation damage was assessed according to the ThioBarbituric Acid Reactive Species (TBARS) method [85], with malonaldehyde as an end product. Final samples were deposited on a black 96-well plate in duplicates, and absorbance was read every minute for 3 min at 533 nm (emission) and 515 nm (excitation). The standard curve was prepared with known concentrations of tetrametoxypropane.

### 4.6. Carbon, Nitrogen, Phosphorus Content

Six nubbins per species and condition were flash-frozen in liquid nitrogen and stored at −80 °C for later analysis of their CNP content. They were freeze-dried before analysis. In each condition, two nubbins did not have enough tissue to allow for accurate results, thus statistical analyses were performed on four nubbins.

#### 4.6.1. Phosphorus Content

Sub-samples from each freeze-dried nubbin were weighed and digested in a persulfate potassium solution autoclaved at 121 °C [86]. Samples were then placed in triplicate on a 96-well microplate with a solution of ammonium molybdate and incubated at room temperature for 10 min before the addition of a solution of malachite green. After 30 min, the absorbance was measured at 630 nm using a spectrofluorometer (Xenius^®^, SAFAS, Monaco) to quantify the orthophosphate content. The standard curve was prepared with known concentrations of potassium phosphate.

#### 4.6.2. Carbon and Nitrogen Content

Sub samples from each freeze-dried nubbin (1 mg) were placed in a tin cap and analyzed with an EA-IRMS Integra2 (Sercon, Crewe, UK).

### 4.7. Statistical Analyses

Statistical tests were performed using the free, open-source software R. Data were checked for homoskedasticity using Levene’s test on the residual from the linear model [87] and for normality using the Shapiro-Wilk Test. One-way ANOVA tests were used to test for the statistical differences in each species between the enrichment conditions: No enrichment (C), nitrate enrichment (N), phosphate enrichment (P) and combined nitrate and phosphate enrichment (NP). If the assumptions of normality and homoskedasticity were not validated, the non-parametric Kruskal–Wallis test was performed. Analysis was followed by a post hoc test (pairwise comparison with "fdr” correction) when factors’ effects were significant. The statistical outcomes are recorded in a table in the Appendix A. Differences were considered to be significant when the calculated *p* value was below 0.05. 

## 5. Conclusions

This study brings key information on the response of corals to nutrient enrichment. The results first show that species with low nutrient uptake rates might be less responsive to changes in seawater DIN:DIP ratio than species with high uptake rates of nutrients. In these later species, changes in seawater DIN:DIP ratio may induce large changes in elemental composition and an imbalanced N:P ratio in coral tissue. All together, these changes may lead to potentially detrimental effects on coral physiology due to P limitation and, in combination with thermal stress, can exacerbate bleaching [18,88]. High nitrate and low phosphate concentrations are found in coastal reefs subjected to river and/or agricultural and effluent discharges [89,90,91]. Although only one soft coral and one scleractinian coral species were tested in this study, soft corals appear to be more resistant to a change in seawater nutrient concentrations than scleractinian corals. This is due to their lower ability to take up and assimilate dissolved nutrients in seawater [66], as well as their higher tissue C and N content per AFDW. However, this needs further investigation as species-specific differences between scleractinian corals have also been observed [41]. Such differential responses to an unbalanced DIN:DIP ratio may help to understand the differential bleaching susceptibility of corals during thermal stress. Therefore, corals with a higher ability to control their internal N:P ratio and maintain a balanced N:P ratio may be more resistant to bleaching.

## Figures and Tables

**Figure 1 ijms-24-03119-f001:**
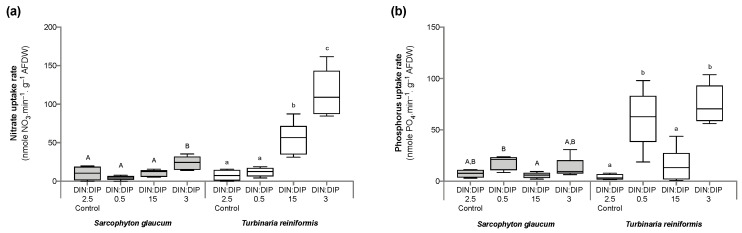
Nutrient uptake rates in the octocoral *Sarcophyton glaucum* and the scleractinian *Turbinaria reniformis*. Boxplots represent the nitrate (**a**) and phosphorus (**b**) uptake rates of *S. glaucum* (grey) and *T. reniformis* (white) under two balanced (DIN:DIP = 2.5 and 3) and two imbalanced (DIN:DIP = 0.5 and 15) ratios. The natural seawater control condition corresponds to a DIN:DIP ratio of 2.5. The box delimits the first and third quartile, the black bar represents the median and the whiskers represent the minimum and maximum values, *n* = 6. Statistically significant differences within species are represented by different letters: uppercase letters (A and B) for *S. glaucum* and lowercase letters (a, b and c) for *T. reniformis*.

**Figure 2 ijms-24-03119-f002:**
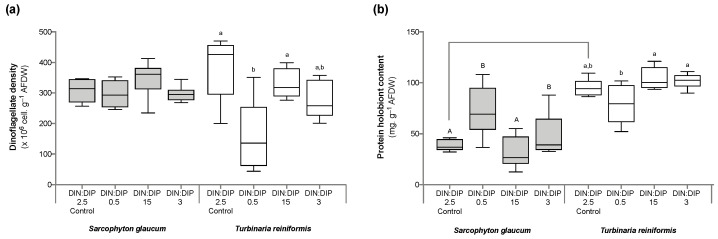
Dinoflagellate density and protein biomass of the octocoral *Sarcophyton glaucum* and the scleractinian *Turbinaria reniformis*. Boxplots represent the symbiotic dinoflagellate density (**a**) and the total protein content (**b**) of *S. glaucum* (grey) and *T. reniformis* (white) under two balanced (DIN:DIP = 2.5 and 3) and two imbalanced (DIN:DIP = 0.5 and 15) ratios. The natural seawater control condition corresponds to a DIN:DIP ratio of 2.5. The box delimits the first and third quartile, the black bar represents the median and the whiskers represent the minimum and maximum values, *n* = 6. Statistically significant differences within species are marked by different letters: uppercase letters (A and B) for *S. glaucum* and lowercase letters (a and b) for *T. reniformis*. Statistically significant differences between species, only for the control condition, are marked by an asterisk.

**Figure 3 ijms-24-03119-f003:**
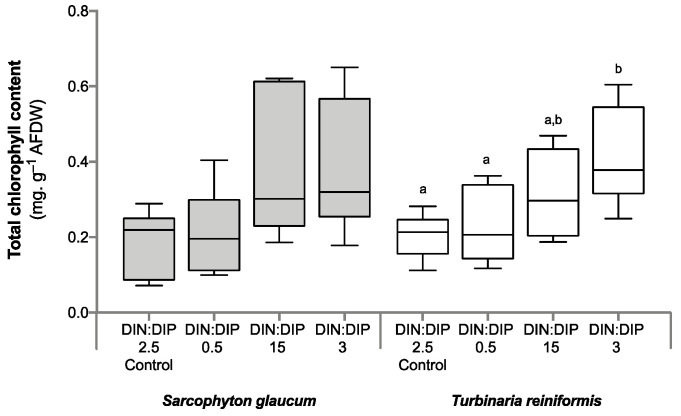
Total chlorophyll a and c_2_ content in the octocoral *Sarcophyton glaucum* and the scleractinian *Turbinaria reniformis*. Boxplots represent the total chlorophyll a and c2 content of *S. glaucum* (grey) and *T. reniformis* (white) under two balanced (DIN:DIP = 2.5 and 3) and two imbalanced (DIN:DIP = 0.5 and 15) ratios. The natural seawater control condition corresponds to a DIN:DIP ratio of 2.5. The box delimits the first and third quartile, the black bar represents the median and the whiskers represent the minimum and maximum values, *n* = 6. Statistically significant differences within species are marked by different letters: lowercase letters (a and b) for *T. reniformis*.

**Figure 4 ijms-24-03119-f004:**
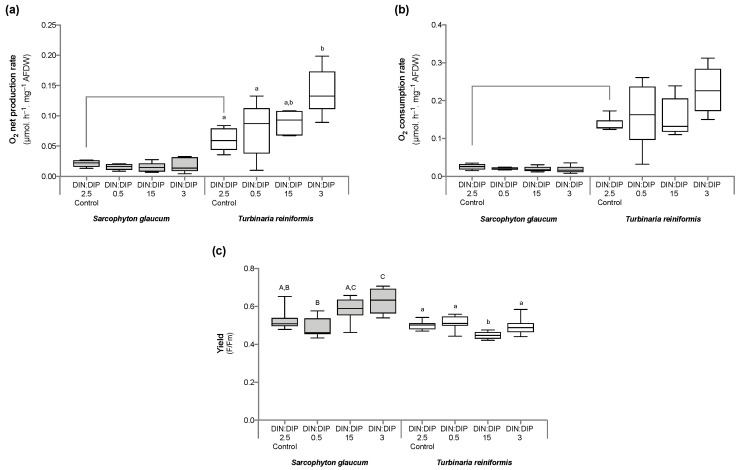
Photosynthetic capabilities in the octocoral Sarcophyton glaucum and the scleractinian Turbinaria reniformis. Boxplots represent the net photosynthetic (**a**) and respiration (**b**) rates and the photosynthetic efficiency of the photosystem II (**c**) of *S. glaucum* (grey) and *T. reniformis* (white) under two balanced (DIN:DIP = 2.5 and 3) and two imbalanced (DIN:DIP = 0.5 and 15) ratios. The natural seawater control condition corresponds to a DIN:DIP ratio of 2.5. The box delimits the first and third quartile, the black bar represents the median and the whiskers represent the minimum and maximum values, *n* = 6. Statistically significant differences within species are marked by different letters: uppercase letters (A, B and C) for *S. glaucum* and lowercase letters (a and b) for *T. reniformis*. Statistically significant differences between species, only for the control condition, are marked by an asterisk.

**Figure 5 ijms-24-03119-f005:**
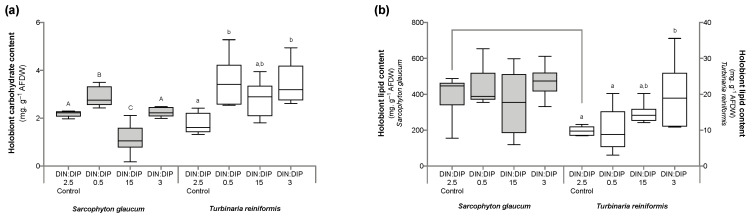
Composition in carbohydrate and lipid in the octocoral *Sarcophyton glaucum* and the scleractinian *Turbinaria reniformis.* Boxplots represent the carbohydrate (**a**) and lipid (**b**) content of *S. glaucum* (grey) and *T. reniformis* (white) under two balanced (DIN:DIP = 2.5 and 3) and two imbalanced (DIN:DIP = 0.5 and 15) ratios. The natural seawater control condition corresponds to a DIN:DIP ratio of 2.5. The box delimits the first and third quartile, the black bar represents the median and the whiskers represent the minimum and maximum values, *n* = 6. Statistically significant differences within species are marked by different letters: uppercase letters (A, B and C) for *S. glaucum* and lowercase letters (a and b) for *T. reniformis*. Statistically significant differences between species, only for the control condition, are marked by an asterisk.

**Figure 6 ijms-24-03119-f006:**
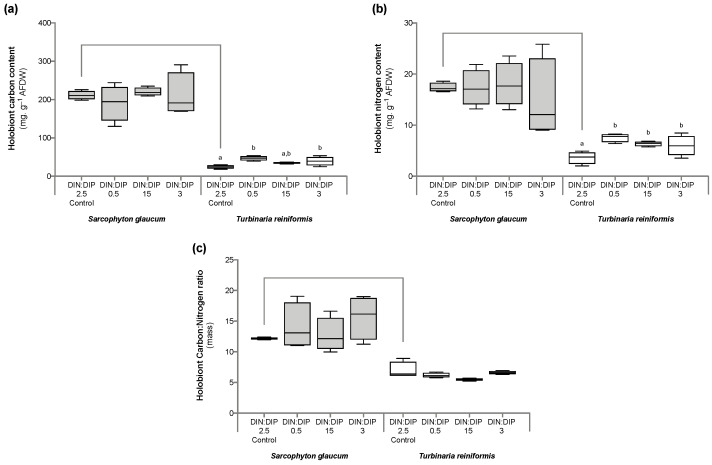
Composition in carbon, nitrogen and the carbon-to-nitrogen (C:N) ratio in the octocoral *Sarcophyton glaucum* and the scleractinian *Turbinaria reniformis*. Boxplots represent the carbon (**a**) and nitrogen (**b**) contents and the C:N ratio (**c**) of *S. glaucum* (grey) and *T. reniformis* (white) under two balanced (DIN:DIP = 2.5 and 3) and two imbalanced (DIN:DIP = 0.5 and 15) ratios. The natural seawater control condition corresponds to a DIN:DIP ratio of 2.5. The box delimits the first and third quartile, the black bar represents the median and the whiskers represent the minimum and maximum values, *n* = 6. Statistically significant differences within species are marked by different letters: lowercase letters (a and b) for *T. reniformis*. Statistically significant differences between species, only for the control condition, are marked by an asterisk.

**Figure 7 ijms-24-03119-f007:**
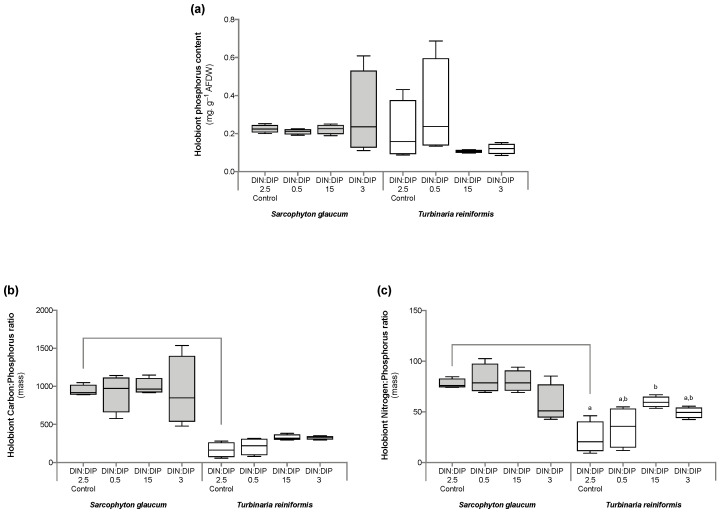
Composition in phosphorus and the carbon-to-phosphorus (C:N) and nitrogen-to-phosphorus (C:P) ratios in the octocoral *Sarcophyton glaucum* and the scleractinian *Turbinaria reniformis.* Boxplots represent the phosphorus (**a**) content and the C:P (**b**) and N:P (**c**) ratios of *S. glaucum* (grey) and *T. reniformis* (white) under two balanced (DIN:DIP = 2.5 and 3) and two imbalanced (DIN:DIP = 0.5 and 15) ratios. The natural seawater control condition corresponds to a DIN:DIP ratio of 2.5. The box delimits the first and third quartile, the black bar represents the median and the whiskers represent the minimum and maximum values, *n* = 6. Statistically significant differences within species are marked by different letters: lowercase letters (a and b) for *T. reniformis*. Statistically significant differences between species, only for the control condition, are marked by an asterisk.

## Data Availability

Data are available upon request.

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
