# Peer review of "Species-Specific Response of Corals to Imbalanced Ratios of Inorganic Nutrients"

_ijms, 2023, doi:10.3390/ijms24043119_

Round 1
Reviewer 1 Report
The manuscript presented by Blanckaert is an important contribution to understanding the physiological response of two poorly studied coral species. The stressor associated with the study is considered one of the main threats (in addition to the thermal stressor), and more importantly, one that, with proper management actions, can be controlled (considering the urban or human inputs). However, there are some concerns regarding the manuscript that should be addressed before publication. The authors use two species with different physiology, since one is a coral species characterized as "hard" and the other "soft", so it is not clear what the authors expected if it is known in advance that their dependence and nutrient fixation capacity will be different. It is important to clarify and, above all, be conservative and not overreach during the comparisons between species. Likewise, it is not clear why they used the levels of nutrients described in the study; Although it is a manuscript focused on physiology, it is important to understand what are the current conditions to which coral species are subjected, throughout the year (and even between years), and if the conditions described can be considered as stressful or normal within the site. And if they are stressful, on what basis was the increase considered? measurements from other nearby sites?. Also, please check all the typo mistakes (italics in the species names, as also units) which are consistently detected.
Line 9. Growth is an important parameter to determine health or physiology; however, this is not a parameter used in the present manuscript, I would suggest changing growth for changes in physiology or physiological response.
Line 11. I understand within the introduction and materials and methods section why they use the term imbalanced; however, any section describes which are balanced vs. unbalanced nutrient conditions, so how can the authors attribute this term along the manuscript.
Line 21. more affected, or have a differential response?
Line 27. How much / less are low levels?
Line 31. suggestion change to convert to metabolize.
Line 31. suggestion change passed for translocated.
Line 34. All the manuscript is focused in the endosymbiont-host response. However, bacteria also contribute with the use of inorganic nutrients (and, more importantly, during bleaching events), it would be important to include it in a few sentences as part of the introduction.
Line 40-41. It can not be considered as the same stressors natural (upwellings) vs anthropogenic stressors (urban wastes); this idea should be rewritten.
Line 59-60. It is confusing, as the authors hypothesize the negative effect of the nutrient input, however, they describe the positive effect on the coral physiology. Please clarify the hypothesis.
Line 84. This is the first appearance of NP condition, so please clarify (it is later observed in the materials section).
Lines 84-85. The use of a supplementary table for the statistical results can be used to avoid over-text, however, is essential to include the data on statistical differences in the main text.
Line 101. Figure 1. It is confusing the use of letters for statistical differences. both species have a and b, even when the analysis was separate, the suggestion is to use different letters per species differences
Line 120. As presented, figure 3 is confusing. The "colors" and "stripes" are not good enough to see the differences.
General comment in figures. Within the present form is not possible to observe the boxes with low values of media and error; also, the arrangement is confusing. As a suggestion, use a single table, where all parameters can be included for both species (or use a single table per species)
General comment in the result section. The present form is confusing as the authors describe first the variation in biomarkers, and then the contribution of the condition to that fluctuation/change. So is not clear what attributes the differences are, considering that no statistical data is included in the main text.
Discussion section. The authors argue a differential response between both species. However, when considering each of the parameters, the control was different for each species; since no statistical analysis was carried out comparing species, how can the authors argue that it was different? This argument is based on the individual analysis of each species and the analysis of each species vs. each of the conditions/parameters.
Also I suggest rather than a comparison among species, explain how hard vs soft corals' physiology cope with these stressors. In this section, the relevance of approaching the coral as a holobiont of animal + microalga + bacteria increases, since some parameters, such as lipids, are described at the holobiont level, and the differences between hard and soft coral can be attributed to the role of bacteria; although it was not a considered parameter, it should be discussed. Finally, this leads me to the fact that S. glaucum was not as affected by stressors as the hermatypic species, however, I would like to go back tothe previous argument, is this typical of the physiology of a soft coral? so what would be the stressful conditions for this one? or the parameters used cannot be the same for both spectra of organisms? The results have high scientific relevance, however, I do not consider it appropriate to establish the stress level of two species that due to their physiology, will respond differently as they have different metabolic paths. By discussing separately at the species level, instead of at the marker level, it can be concluded which species will possibly dominate or will be most affected during nitrogen imbalance conditions, with this having an implication at the level of ecosystem functionality.
Lines 325-333. The experimental design is clear, But based on what are the acclimatization conditions considered? those were the in situ conditions?? Or why is it considered an adequate basal condition?
Lines 331-344. How did they consider the nutrient ratios? Reports from that site or from nearby places? Or based on some report of future scenarios?.
Also as a general comment, why were biomarkers assessed at the beginning and at the end of the treatment? In the case of lipids, it is understood that the response is slower, however, for photosynthetic efficiency (and for which it is not necessary to sacrifice the nubbin) it would have been important to evaluate in shorter periods of time. Also please, along with the materials and method section, check all the units and typo mistakes.
Lines 370. Is not clear whether different nubbins were used for each analysis, however it is unclear how they were used for each marker
Line 403. Suggestion change 6 for six.
Reviewer 2 Report
This manuscript entitled with “Species-specific response of corals to imbalanced ratios of inorganic nutrients” provides valuable information to understand the role of inorganic nutrient to coral symbioses. Unfortunately, authors did not pay attention to explain the impact of balance and imbalance of DIN/DIP on the interactions between coral host and algal symbiont. I will encourage author use a section to discuss this part. Below are comments and questions.
Major comments
The writing of manuscript requires tidy-up. Many places are not clear or not in the correct format. For example, in method section, authors used 12 colonies to produce 8 nubbins from each colony and separated them into 8 aquariums, which meant totally 12 nubbins from different colony were put in an aquarium. But 8 aquariums were divided into 4 different treatments, which meant authors considered tank effect. When authors said they used 6 nubbins for each treatment condition, I don’t know how would authors select the nubbin from two separate aquariums to produce replication. According to the experimental design, 6 nubbins for photosynthesis related analysis, 6 nubbins for tissue composition, 6 nubbins for NETAC, 6 nubbins for LPO, 4 nubbins for element analysis, but authors only have 2 aquariums * 12 colonies = 24 nubbins for each treatment condition. Authors shall describe in detail how would they use these nubbins to run so many tests. There are also not clear in Figure. For example, Figure 1 described the uptake rate of nitrate and phosphate (or phosphorus ???) by Turbinaria, but authors did not mention how to measure the uptake rate, instead, authors only mentioned they incubate the coral in different nutrient condition for 5 weeks. I think it is impossible to measure uptake rate by using so long time scale. Also, what does the horizontal line in Fig.1a, 3a, 4b, and 5 mean?
Minor comments and questions
L36-44: There are 4 times of using “on the other hand”. Authors are suggested to rephrase this part to make the interpretation more logically.
L326: Sarcophyton glaucum and Turbinaria reniformis shall be in Italic format.
L331, 360: m−2 s−1 shall use superscript.
L360-361: O2, subscript shall be used
L333: Artemia salina shall be in Italic format.
L381: Q1: why did authors use freeze-dried tissue slurry for symbiont counting? How to make sure the cell integrity, especially no added with microscopic observation?
Q2: I am not sure if freeze drying could protect chl. a not to be decomposed. I am afraid the chl. a was underestimated.
L402: Q1: lipid of holobiont was measured with a freeze-dried nubbin? That meat the sample included skeleton. Why not use AFDW? If the sample included skeleton, then the process of standardization of lipid content will have unexpected error.
L414: Q1: Same problem raised as lipid measurement. How did exactly the sample preparation for CHO content measurement.
Round 2
Reviewer 2 Report
I have no further question, except one minor concern about the extraction of lipid. Lipid extraction from coral tissue slurry is very difficult to get correct results, because very difficult to recovery all tissue slurry after blastng of tissue.
Author Response
I have no further question, except one minor concern about the extraction of lipid. Lipid extraction from coral tissue slurry is very difficult to get correct results, because very difficult to recovery all tissue slurry after blastng of tissue.
Authors: We agree that tissue extraction is not always optimal, andsome tissue loss may occur. However, as the lipid measurements were normalised by ash-free dry weight, the bias due to tissue extraction is reduced. Therefore, our results should befairly accurate.
Note: We updated the Data availability and Acknowledgments statement in the manuscript.